

# Technical note: A theoretical study on the mechanism of citric acid-driven multi-component nucleation of sulfuric acid-base-water clusters

Xiaoli Gong[1,2], Liyao Zhu[1], Renyi Zhang[3]

[1]Electronics and Information College, Hangzhou Dianzi University, Hangzhou 310018, China
[2]Institute of Carbon Neutrality and New Energy, Hangzhou Dianzi University, Hangzhou 310018, China
[3]Department of Atmospheric Sciences and Department of Chemistry, Texas A&M University, College Station, TX 77843, USA.

*Correspondence to*: Xiaoli Gong (gxl@hdu.edu.cn) and Renyi Zhang (renyi-zhang@geos.tamu.edu)

**Abstract.** New particle formation (NPF) is one of the important sources of aerosol and an important reason for the rapid increase of $PM_{2.5}$ mass concentration in polluted areas. It has been shown that citric acid, present in atmosphere, is a potential precursor of new particles and may play a role in the nucleation process of new particles. However, the exact mechanism by which citric acid contributes to nucleation remain unclear. The thermodynamically stable geometry of $SA \cdot AM \cdot W_n \cdot CA_m$, $SA \cdot DMA \cdot W_n \cdot CA_m$, and $SA \cdot AM \cdot DMA \cdot W_n \cdot CA_m$ (n = 0 - 4, m=0-1) clusters were optimized and the Gibbs free energy and

hydration distribution were calculated at the M06-2X/6-311+G(2d, p) level in this study. The results demonstrate that three carboxyl groups (-COOH) and one hydroxyl group (-OH) of citric acid can act as both hydrogen donors and acceptors through hydrogen bonding interactions with sulfuric acid-base-water clusters. This interaction lowers the nucleation barriers ($\triangle G <$ 0), indicating an energetically favorable reaction. At three relative humidities (RH), anhydrous and monohydrate forms dominate in $SA \cdot AM \cdot W_n \cdot CA$ clusters, as well as $SA \cdot AM \cdot DMA \cdot W_n \cdot CA$ (n = 0 - 4) clusters; while anhydrous form dominates

in $SA \cdot DMA \cdot W_n \cdot CA$ (n = 0 - 4) clusters. These findings suggest that citric acid reduces the hydrophilicity of these clusters. In conclusion, the involvement of citric acid in atmospheric processes is conductive to cluster formation, thereby facilitating the multicomponent nucleation of sulfuric acid-base-water clusters. Overall, our study highlights how citric acid participates in and enhances new particle formation processes.

## 1 Introduction

New particle formation (NPF) represents a microscale evolutionary process from molecules to nanoparticles, wherein gas-phase vapors originating from biological or anthropogenic sources first become supersaturated and subsequently condense into thermodynamically stable clusters. These clusters then serve as critical nuclei for further growth through condensation with other gas-phase vapors or by inter-particle collision (Zhang et al., 2012), which significantly contributes to the rapid increase in atmospheric $PM_{2.5}$ mass concentration observed in polluted regions. Currently, new particle formation events are frequently

observed across various atmospheric environments worldwide, and understanding the nucleation mechanism of new particles



in the atmosphere, particularly the thermodynamic stability of initial molecular clusters, has emerged as a prominent research topic within international atmospheric chemistry (Yao et al., 2018).

    The molecular level process of nucleation, which topically involves the formation of stable molecular clusters through hydrogen bonding of gas-phase nucleating precursors, is crucial for comprehending the mechanism behind new particle

formation (Zhang, 2010). Nucleation is evident in the creation of thermodynamically stable molecular clusters via condensation of low volatile gases. It is influenced by factors such as solar radiation, intensity, ambient temperature and relative humidity. New particles exhibit a wide range of shapes and sizes, while the key gaseous precursors involved in their formation vary significantly across different environments and geographic locations, resulting in complex chemical compositions with noticeable seasonal and geographical variations. Numerous studies have demonstrated that sulfuric acid plays an important

role as a precursor in nucleation processes (Kulmala et al., 2000; Sipilä et al.,2010); Doyle was the first to propose the binary homogeneous nucleation theory involving sulfuric acid-water through numerical calculations (Doyle et al., 1961). Ammonia is a prevalent substance in the atmosphere, whereas amines are derivatives of ammonia with stronger alkalinity than ammonia itself; they can form stable complexes or salts with sulfuric acid or organic acids. Consequently, ternary homogeneous nucleation theories involving sulfuric acid-water-ammonia (Kirkby et al., 2011) and sulfuric acid-water-organic amines

(Almeida et al., 2013) have been proposed; however, these mechanisms fail to fully explain new particle formation events occurring within atmospheric boundary layers or across different regions worldwide. As research progressed further, it became apparent that actual atmospheric nucleation processes were much more intricate and Zhang et al. first proposed the involvement of organic acids in promoting nucleation (Zhang et al., 2004), i.e., sulfuric acid nucleation was significantly enhanced in the presence of organic acids such as benzoic acid. Subsequently, a series of theoretical studies have been carried out to understand

how carboxylic acids stabilize clusters, ranging from low molecular weight monocarboxylic acids (e.g., formic acid (Liu et al., 2021), glycolic acid (Zhang et al., 2017) and lactic acid (Li et al., 2017)) to complex bicarboxylic and tricarboxylic acids containing multiple functional groups (e.g., oxalic acid (Miao et al., 2015; Yang et al.,2021), pinic acid (Elm et al., 2014), *cis*-pinonic acid (Zhang et al., 2009), benzoic acid (Schnitzler et al., 2014), malonic acid (Zhang et al., 2018; Wang et al.,2021), succinic acid (Lin et al., 2019; Xu et al.,2013; Wang et al.,2021), malic acid (Liu et al., 2022), tartaric acid (Wang et al., 2023),

and 3-methyl-1,2,3-butanetricarboxylic acid (Zhang et al., 2018; Xia et al., 2021)). However, due to the complexity and diversity of functional groups contained in atmospheric organic acids, the interactions between organic acids and nucleation precursors remain inconclusive. Therefore, ternary organic acids (e.g., citric acid) with multifunctional groups (e.g., hydroxyl and carboxyl functional groups) will be the target of this paper.

    Citric acid ($C_6H_8O_7$) is a tricarboxylic acid containing one hydroxyl group,which have been detected in sea salt particles

along the Atlantic coast, being present in atmospheric particles (Fu et al., 2008) and may be directly emitted from tangerine is likely adsorbed on pollens emitted from Japanese cedar (Jung et al., 2011), suggesting that citric acid may be involved in new particle formation, and thus citric acid is a potential precursor for new particles. The higher concentration, lower volatility, lower vapor pressure, higher water solubility, and higher O:C ratio of citric acid (Liu et al., 2012; Shi et al., 2017; Lv et al., 2020; Wyrzykowski et al., 2011; Chim et al., 2018) is one of highly oxygenated multifunctional organic molecules (HOMs)





in the atmosphere, suggesting that citric acid could be involved in the nucleation process of new particle formation. The non-homogeneous absorption of amines by citric acid (Liu et al., 2012) and the volatility and hygroscopicity of internally-mixed particles consisting of citric acid, ammonium sulphate, and water were investigated in laboratory (Shi et al., 2017; Lv et al., 2020). But whether citric acid, as a potential precursor, plays any role in new particle nucleation remains an unanswered question.

Quantum chemical calculations on nucleating molecular clusters are one of the important research tools for studying new particle nucleation at present. Therefore, the $SA \cdot AM \cdot W_n \cdot CA_m$, $SA \cdot DMA \cdot W_n \cdot CA_m$, and $SA \cdot AM \cdot DMA \cdot W_n \cdot CA_m$ (n = 0 - 4, m=0-1) clusters are the object in the study, which are formed from citric acid (CA), sulfuric acid (SA), ammonia (AM), dimethylamine (DMA) and water (W) on the basis of a full-domain potential energy surface search algorithm combined with a high-precision quantum chemical calculation method and the following three calculations are performed at the M06-2X/6-

311+G(2d,p) level in this paper: (1) Optimization of the thermodynamically most stable structures of the above three clusters; (2) Calculation of thermodynamic properties such as vibrational frequencies and reaction Gibbs free energies for the thermodynamically most stable structures of the above three clusters; (3) Calculate the hydration distribution of the most stable structures of the above three clusters and the main forms of existence at three common atmospheric humidities, i.e., RH=20%, 50% and 80%.

Through the above three calculations, the aim of this paper is to understand whether citric acid participates in and promotes the formation of sulfuric acid-base-water clusters and the nucleation mechanism of the above clusters at the molecular level.

## 2 Calculation methods

Quantum chemical calculations on nucleating molecular clusters are one of the important research tools for studying new

particle nucleation at present (Elm et al., 2020). In this study, we first perform structure search and optimization to obtain the thermodynamically most stable structures of $SA \cdot AM \cdot W_n \cdot CA_m$, $SA \cdot DMA \cdot W_n \cdot CA_m$, and $SA \cdot AM \cdot DMA \cdot W_n \cdot CA_m$ (n = 0 - 4, m=0-1) clusters; and then thermodynamic calculations are carried out to get the thermodynamic properties such as vibrational frequencies and reaction Gibbs free energies of the above three clusters; and finally, the calculation of hydration distribution.

### 2.1 Structure search and optimization

Firstly, the genmer tool in the Molclus package (Lu, 2019) was used to randomly generate 1000 possible initial structural configurations and then the 100 lowest energy structures were got by optimizing the above 1000 structures using the PM6 semi-empirical method. Secondly, 30 lowest-energy configurations were got by re-optimizing the above 100 structures using the Gaussian 09 package (Frisch et al., 2009) using the M06/6-31G* level. Finally, the M06/6-311+G(2d, p) level was used to optimize again and compute the frequencies of the above 30 lowest-energy configurations to obtain the global lowest energy

stable configuration with no imaginary frequencies on the potential energy surface.





Many computational studies have shown that the M06-2X functional is more reliable than other common functional, such as B3LYP, PW91, and B3RICC2 in obtaining stable geometrical configurations of small clusters and reactive Gibbs free energies (Liu et al., 2021; Zhang et al., 2017; Li et al., 2017; Miao et al., 2015; Yang et al.,2021; Elm et al., 2014; Zhang et al., 2009; Schnitzler et al., 2014; Zhang et al., 2018; Wang et al.,2021; Lin et al., 2019; Xu et al.,2013; Wang et al., 2021; Liu et al., 2022; Wang et al., 2023; Zhang et al., 2018; Xia et al., 2021; Elm et al., 2020). Table 1 lists the calculated results of this study as well as previous theoretical and experimental results for the reaction Gibbs free energy of the basic atmospheric clusters. As can be seen from Table 1, taking into account the computational accuracy, cost-effective and contribution to the free energy as well as the arithmetic, the M06-2X/6-311+G (2d, p) level was chosen for the paper.

Table 1: Theoretical and experimental values of the free energy change for several basic reactions in kcal/mol.

| Rection | This study | Refs | | | Refs |
|---|---|---|---|---|---|
| | M06-2X/6-311+G (2d, p) | PW91PW91 /6-311++G (2df, 2pd)[21] | M06-2X/6-311++G (3df, 3pd)[21] | ωB97X-D/6-311++G (3df, 3pd)[25] | |
| SA+AM→SA·AM | -7.98 | -7.65 | -8.00 | -6.20 | -8.5[38]*, -7.77[39], -6.64[40], -7.84[41] |
| SA+DMA→SA·DMA | -11.22 | -11.13 | -11.24 | -11.99 | -13.66[40], -11.38[42] |
| SA·AM+W→SA·AM·W | -2.64 | -1.48 | -0.07 | | -1.41[39], -1.67[43] |
| SA·DMA+W→SA·DMA·W | -4.85 | -3.06 | -3.63 | | -3.67[42], -2.89[43] |

*Corresponds to experimental results.

## 2.2 Thermodynamic calculations

The zero-point energy (ZPE) is the vibrational energy of the cluster at 0K. The frequency is the quadratic numerical difference of the energy. ZPE and frequency calculations are necessary to obtain the thermodynamic parameters of the clusters. It's worth noting that the computational levels (method and basis set) for frequency calculation and geometry optimization need to be consistent.

Thermodynamic properties such as the binding energy ($\triangle E$), zero-point energy (ZPE), formation enthalpy ($\triangle H$), and Gibbs free energy ($\triangle G$) were calculated at 298.15 K and 1 atmosphere. In order to obtain a global minimum structure on the potential energy surface, which is a steady state, the vibrational frequencies and infrared spectra of the clusters were calculated in the paper using the M06-2X/6-311+G (2d, p) level of theory. Dispersion corrections being applied, the integration lattice points are ultrafine, and the convergence limit is tight.

For a stepwise hydration process of a cluster, the chemical reactions are as follows:

$$[\text{cluster}]\cdot W_{n-1}+W\rightarrow[\text{cluster}]\cdot W_n , \tag{R1}$$

The stepwise hydration Gibbs free energies $\triangle G_{\text{step}}$ of the reaction is given by the following equation:



$$\triangle G_{step} = G_{[cluster]\cdot W_n} - (G_{[cluster]\cdot W_{n-1}} + G_W) , \qquad (1)$$

When a CA molecule is added to the cluster, the reaction is as follows:

[cluster]+CA→[cluster]·CA,                                               (R2)

Then the reactive Gibbs free energy $\triangle G_{react}$ is given by the following equation:

$$\triangle G_{react} = G_{[cluster]\cdot CA} - (G_{[cluster]} + G_{CA}) , \qquad (2)$$

For a chemical reaction in which a number of molecules form a cluster:

$aA + bB + cC \rightarrow A_a\cdot B_b\cdot C_c,$                                 (R3)

Then the reactive Gibbs free energy $\triangle G_{react}$ of the cluster is given by the following equation:

$$\triangle G_{react} = G_{A_a\cdot B_b\cdot C_c} - (aG_A + bG_B + cG_C) , \qquad (3)$$

## 2.3 Hydration distribution

The hydration (the hydrophilicity) of the clusters, is affected by the humidity in the atmosphere, which may be have an impact

on the mechanism of new particle nucleation involving citric acid in the real atmosphere. Therefore, hydration distribution of clusters was calculated at different relative humidity levels. According to the hydration distribution calculated by Henschel *et al.* [43], the relative hydrate population $x_n$ of a hydrate containing $n$ water molecules is as shown below:

$$x_n = \left(\frac{p_{(H_2O)}}{p^0}\right)^n x_0 e^{\frac{-\Delta G_n}{RT}} , \qquad (4)$$

where $p_{(H_2O)}$ is the partial pressure of water, $p^0$ is the reference partial pressure of water at 1 atm, $x_0$ is the population of core

cluster, $\Delta G_n$ is the total Gibbs free energy of hydration, $R$ is the molar gas constant (8.314 J/mol·K), and $T$ is the standard temperature (298.15 K).

The formula for $p_{(H_2O)}$ is shown below:

$$p_{(H_2O)} = p_{(H_2O)eq}\times RH , \qquad (5)$$

where $p_{(H_2O)eq}$ is the saturated vapor pressure of water as a function of temperature, and RH is the relative humidity.

## 3 Results and discussion

We will discuss the optimized global minimum energy structures on the potential energy surface, thermodynamical properties, such as the binding energy ($\triangle E$), zero-point energy (ZPE), formation enthalpy ($\triangle H$), and Gibbs free energy of the reaction($\triangle G$), and hydration distribution in terms of $G$ at $T$ = 298.15 K and $p$ = 1 atm of SA·AM·$W_n$·CA$_m$, SA·DMA·$W_n$·CA$_m$, and



SA·AM·DMA·W$_n$·CA$_m$ (n = 0 - 4, m=0-1) clusters, in order to show the mechanism of citric acid-driven multicomponent
nucleation of sulfuric acid-base-water clusters in this section.

### 3.1 Optimized global minimum energy structures

The length and strength of the hydrogen bond as well as the number of proton transfers play an important role in stabilizing
the clusters. The shorter (stronger) the hydrogen bond and the higher the number of proton transfers, the more stable the cluster
is.   Thus,   the   optimized   global   minimum   energy   structures   of   SA·AM·W$_n$·CA$_m$,   SA·DMA·W$_n$·CA$_m$   and
SA·AM·DMA·W$_n$·CA$_m$ (n = 0 - 4, m=0-1) clusters are discussed from two aspects, 1) the length and strength of hydrogen
bond; 2) the number of proton transfer.

### 3.1.1 Optimized global minimum energy structures of SA·AM·W$_n$·CA$_m$ (n = 0 - 4, m=0-1) clusters

The optimized global minimum energy structures for SA·AM·W$_n$·CA$_m$ (n = 0 - 4, m=0-1) clusters are shown in Fig. 1. The
number and length of hydrogen bonds (in Å), the number of proton transfer, and the contribution of CA in the formation of
hydrogen bonds in all clusters are listed in Table S1-S3 in SI respectively.

As shown in Fig. 1, the SA·AM cluster contains a hydrogen bond with a bond length of 1.539 Å. After the addition of
one water, the SA·AM·W cluster contains a strong hydrogen bond, a medium hydrogen bond, and a relatively weak hydrogen
bond with bond lengths of 1.563 Å, 1.719 Å, and 2.067 Å respectively. But no proton transfer occurs in any of the two clusters.
The SA·AM·W$_2$, SA·AM·W$_3$, and SA·AM·W$_4$ clusters all form a cage-like structure containing five (four medium and one
relatively weaker strength), six (four medium and two relatively weaker strength), and seven (one strong, four medium and
two relatively weaker strength) hydrogen bonds, respectively, and all undergo a proton transfer phenomenon in which the
proton is transferred from SA to AM to form the NH$_4^+$ ion and the HSO$_4^-$ ion.

When a citric acid molecule is added to SA·AM·W$_n$ (n = 0 - 4) to form SA·AM·CA·W$_n$ (n = 0 - 4) clusters, the number
of hydrogen bonds within the clusters increases by 3 to 5, and the lengths and numbers of the hydrogen bonds are shown in
Table S1. The optimized global minimum energy configurations of the SA·AM·CA·W$_n$ (n = 0 - 4) clusters are all cage-like
structures and all undergo a proton transfer phenomenon, where the proton is transferred from SA to AM to form NH$_4^+$ ion
and the HSO$_4^-$ ion. One alpha-carboxyl group (α-COOH) and one beta-carboxyl group (β-COOH) of CA are involved in the
formation of three hydrogen bonds in the SA·AM·CA·W$_n$ (n = 0, 2, 4) cluster, and four hydrogen bonds in the SA·AM·CA·W
cluster, while not only one α-carboxy group and one beta-carboxy group but also one beta-hydroxy group (β-OH) of CA are
involved in the formation of four hydrogen bonds in the SA·AM·CA·W$_3$ cluster.



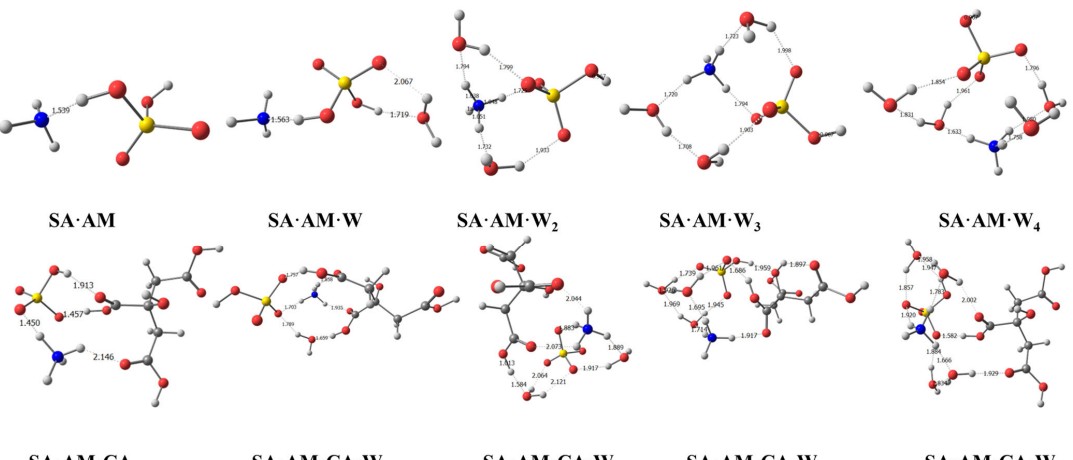

**Figure 1: The optimized global minimum energy structures of SA·AM·Wn (n = 0 - 4) clusters and SA·AM·CA·Wn (n = 0 - 4) clusters on the potential energy surface. Sulfur, carbon, oxygen, nitrogen, and hydrogen atoms are represented by yellow, gray, red, blue, and white balls, respectively. Hydrogen bonds are indicated by dashed lines.**

### 3.1.2 Optimized global minimum energy structures of SA·DMA·Wₙ·CAₘ (n = 0 - 4, m=0-1) clusters

As shown in Fig. 2, the SA·DMA cluster contains a hydrogen bond with a bond length of 1.392 Å and occurs a proton transfer phenomenon in which the proton is transferred from SA to DMA to form the $(CH_3)_2NH_2^+$ ion and $HSO_4^-$ ion. SA·DMA·Wₙ (n = 1 - 4) clusters all form a cage-like structure containing three (one strong and two medium strength), five (four medium and one relatively weaker strength), six (four medium and two relatively weaker strength), and seven (one strong, four medium and two relatively weaker strength) hydrogen bonds, respectively, and all undergo a proton transfer phenomenon in which the proton is transferred from SA to AM to form the $(CH_3)_2NH_2^+$ ion and $HSO_4^-$ ion.

When a citric acid molecule is added to SA·DMA·Wₙ (n = 0 - 4) to form SA·DMA·CA·Wₙ (n = 0 - 4) clusters, the number of hydrogen bonds within the clusters increases by 1 to 4, and the lengths and numbers of the hydrogen bonds are shown in Table S1. The optimized global minimum energy configurations of the SA·DMA·CA·Wₙ (n = 0 - 4) clusters are all cage-like structures in Fig. 2 and all undergo a proton transfer phenomenon, where the proton is transferred from SA to DMA to form $(CH_3)_2NH_2^+$ ion and $HSO_4^-$ ion.

H on α-COOH and β-COOH act as donors and O as an acceptor while forming two pairs of hydrogen bonds with O on sulfuric acid and H on dimethylamine in the SA·DMA·CA cluster.

H on α-COOH act as donors and O as an acceptor while forming two hydrogen bonds with O on sulfuric acid and H on dimethylamine, meanwhile O on β-OH act as acceptor forming one hydrogen bond with H on water in the SA·DMA·CA·W cluster.



In the SA·DMA·CA·W$_2$ cluster, two α-carboxyl and one β-carboxyl group of CA are involved in the formation of 5 hydrogen bonds. One α-COOH acts as both hydrogen donor and acceptor, forming two hydrogen bonds with water. Another α-COOH only act as hydrogen acceptor, forming a relatively weaker strength hydrogen bonding with another water. The β-

COOH group of CA forms two hydrogen bonds with both sulfuric acid and dimethylamine, acting as a hydrogen donor, forming one strong hydrogen bond with sulfuric acid and as an acceptor, forming a relatively weaker strength hydrogen bond with dimethylamine.

In the SA·DMA·CA·W$_3$ cluster, one α-carboxyl group and one β-carboxyl group of CA are involved in the formation of two pairs of hydrogen bonds. One α-COOH forms one strong strength hydrogen bond as hydrogen donor with water and one

relatively weaker strength hydrogen bond as acceptor with DMA. The β-COOH group of CA acts as both donor and acceptor, forming one strong and one medium strength hydrogen bonds simultaneously with two different water molecules.

In the SA·DMA·CA·W$_4$ cluster, one α-carboxyl group and one β-hydroxyl group of CA are involved in the formation of three hydrogen bonds. One α-COOH groups acted as both donor and acceptor while forming one strong and one medium strength hydrogen bonds with SA. The β-OH of CA acted only as an acceptor and formed one medium strength hydrogen bond

with water.

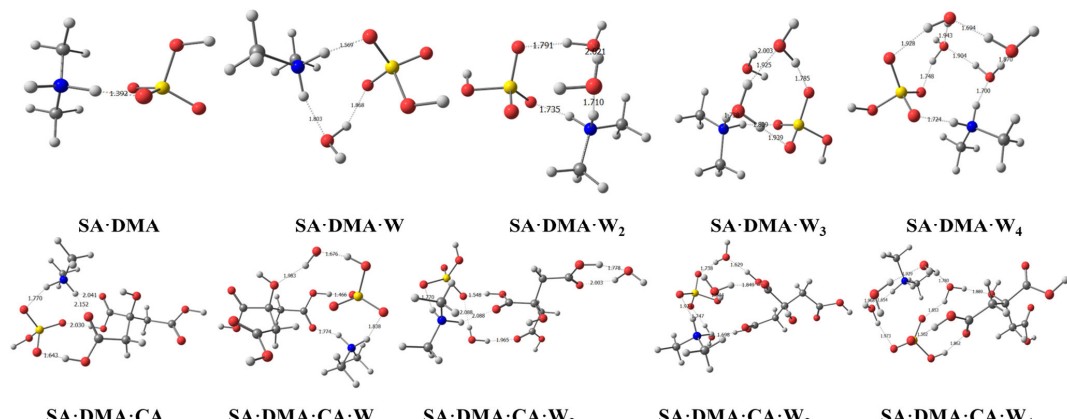

| **SA·DMA** | **SA·DMA·W** | **SA·DMA·W$_2$** | **SA·DMA·W$_3$** | **SA·DMA·W$_4$** |

| **SA·DMA·CA** | **SA·DMA·CA·W** | **SA·DMA·CA·W$_2$** | **SA·DMA·CA·W$_3$** | **SA·DMA·CA·W$_4$** |

**Figure 2: The optimized global minimum energy structures of SA·DMA·W$_n$ (n = 0 - 4) clusters and SA·DMA·CA·W$_n$ (n = 0 - 4) clusters on the potential energy surface. Sulfur, carbon, oxygen, nitrogen, and hydrogen atoms are represented by yellow, gray, red, blue, and white balls, respectively. Hydrogen bonds are indicated by dashed lines.**

**3.1.3 The optimized global minimum energy structures of SA·AM·DMA·W$_n$·CA$_m$ (n = 0 - 4, m=0-1) clusters**

As shown in Fig. 3, the SA·AM·DMA·W$_n$ (n = 0 - 4) cluster contains 2, 4, 5, 7, and 8 hydrogen bonds, respectively, and occurs one proton transfer to form the $(CH_3)_2NH_2^+$ ion and $HSO_4^-$ ion. When a citric acid molecule was added to SA·AM·DMA·W$_n$ (n = 0 - 4) to form the SA·AM·DMA·CA·W$_n$ (n = 0 - 4) cluster, there were 4, 6, 9, 10, and 8 hydrogen bonds within the cluster. Interestingly, there are two protons being transferred within SA·AM·DMA·CA·W$_2$ to form the $HSO_4^-$ ion, $(CH_3)_2NH_2^+$ ion,




NH$_4^+$ ion and R-COO$^-$ ion (R=-CH$_2$-C(OH)(COOH)-CH$_2$-COOH), while the other clusters undergo only one proton transfer, forming the (CH$_3$)$_2$NH$_2^+$ ion and HSO$_4^-$ ion.

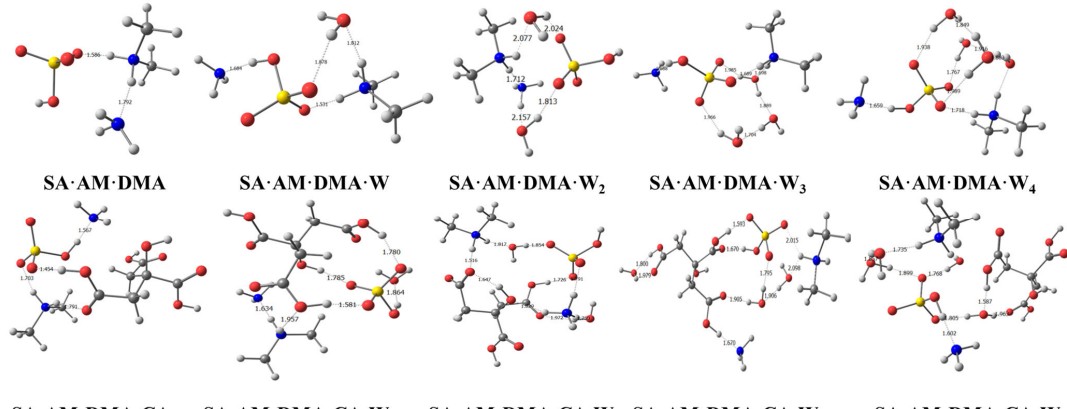

**Figure 3: The optimized global minimum energy structures of SA·AM·DMA·W$_n$ (n = 0 - 4) clusters and SA·AM·DMA·CA·W$_n$ (n = 0 - 4) clusters on the potential energy surface. Sulfur, carbon, oxygen, nitrogen, and hydrogen atoms are represented by yellow, 220 red, blue, and white balls, respectively. Hydrogen bonds are indicated by dashed lines.**

In the SA·AM·DMA·CA cluster, an α-carboxyl group of CA acts as an H donor and acceptor while forming a pair of strong and moderately strong hydrogen bonds with sulfuric acid and dimethylamine.

In the SA·AM·DMA·CA·W cluster, 2 α-carboxyl groups and one hydroxyl group of CA are involved in the formation of 225 4 hydrogen bonds. One α-COOH forms 2 hydrogen bonds with both sulfuric acid and dimethylamine, where CA acts as a hydrogen donor, providing 1 hydrogen to sulfuric acid, while acting as an acceptor, forming another hydrogen bond with dimethylamine. The other α-COOH and β-OH groups of CA both act as donors only, forming 2 hydrogen bonds with the two O atoms of sulfuric acid, respectively.

In the SA·AM·DMA·CA·W$_2$ cluster, two α-COOHs and one β-OH of CA are involved in the formation of five hydrogen 230 bonds. One α-COOH acts only as a hydrogen acceptor and interacts both with dimethylamine and β-OH to form a strong strength intramolecular hydrogen bond and one strong strength intermolecular hydrogen bond, and proton transfer occurs, with H transferring from the α-COOH of CA to dimethylamine to form (CH$_3$)$_2$NH$_2^+$ ions and R-COO$^-$ ions (R=-CH$_2$-C(OH)(COOH)-CH$_2$-COOH). While β-OH is both a hydrogen donor and hydrogen acceptor, it forms a strong and a medium strength hydrogen bond with both α-COOH and ammonia. The other α-COOH acts as both a hydrogen donor and acceptor, 235 interacting with both sulfuric acid and water to form a hydrogen bond that undergoes proton transfer, with H transferring from SA to AM to form HSO$_4^-$ ion and NH$_4^+$ ion.



In the SA·AM·DMA·CA·W$_3$ cluster, three carboxyl groups of CA are all involved in the formation of six hydrogen bonds. An α-COOH simultaneously forms 2 hydrogen bonds with water as an acceptor and ammonia as a donor. The other α-COOH group of CA forming 2 hydrogen bonds with another water molecule not only acts as a donor but also as an acceptor. The β-COOH group of CA forms 2 strong strength hydrogen bonds with sulfuric acid not only as a donor but also as an acceptor.

In the SA·AM·DMA·CA·W$_4$ cluster, both α-carboxyl groups of CA are involved in the formation of one strong and one relatively weaker strength hydrogen bonds. One α-COOH group acts only as a donor and the other α-COOH acts only as an acceptor to form two hydrogen bonds with the same water respectively.

In summary, in terms of structural parameters, such as the length and number of hydrogen bonds within the cluster and the proton transfer phenomenon, the two α-COOH, β-COOH, and β-OH groups of CA, which can act as both hydrogen donors and acceptors, are all involved in the formation of the hydrogen bonds, which reduces the nucleation barrier, and plays an important role in stabilizing the SA·AM·W$_n$, SA·DMA·W$_n$, and SA·AM·DMA·W$_n$ (n = 0 - 4) clusters.

### 3.1 Thermodynamical properties.

The step by step hydration Gibbs free energies for the SA·AM·W$_n$, SA·DMA·W$_n$ and SA·AM·DMA·W$_n$ (n = 0 - 4) clusters are presented in Fig 4 (a) and Tab S4. The reactive Gibbs free energies for the addition of a CA molecule into the SA·AM·W$_n$, SA·DMA·W$_n$ and SA·AM·DMA·W$_n$ (n = 0 - 4) clusters are presented in Fig 4 (b) and Tab S4.

Figure 4(a) shows that at most hydration degrees, the stepwise hydration Gibbs free energy changes are negative, indicating that the hydration are thermodynamically favorable. The fourth hydration (0.3 kcal mol$^{-1}$) of SA·DMA·CA, and the third hydration (1.6 kcal mol$^{-1}$) of SA·AM·DMA·CA are positive. This conclusion coincides with the positive fourth-step hydration energy of SA·DMA·SUA (SUA=succinic acid) and the positive third-step hydration energy of SA·AM·DMA·SUA (Lin et al., 2019), as well as the positive fourth-step hydration energy of SA·DMA·TA (TA=tartaric acid) (Wang et al., 2023).The reason could be that the trihydrate of the SA·DMA cluster with the addition of one carboxylic acid molecule is very stable, whereas the dihydrate of the SA·AM·DMA cluster with the addition of one carboxylic acid molecule is very stable (our calculations have revealed that two proton transfers have taken place in SA·AM·DMA·CA·W$_2$ cluster, which resulted in the formation of HSO$_4^-$ ion, (CH$_3$)$_2$NH$_2^+$ ion, NH$_4^+$ ion, and R-COO$^-$ ion (R=-CH$_2$-C(OH)(COOH)-CH$_2$-COOH)).

Addition of a CA molecule to SA·AM·W$_n$ (n = 0 - 4), the first and the second hydration reaction Gibbs free energies (-8.45 kcal mol$^{-1}$ and -8.52 kcal mol$^{-1}$) are more negative. While addition of a CA molecule to SA·DMA·W$_n$ and SA·AM·DMA·W$_n$ (n = 0 - 4) clusters, only the first hydration reaction Gibbs free energies (-13.45 kcal mol$^{-1}$ and -10.24 kcal mol$^{-1}$) are more negative.

It is clear that citric acid facilitates the stabilization of the clusters in this paper, and furthermore, the values of the Gibbs free energies of the reactions added to the multicomponent clusters are related to the degree of hydration as well as to the precursors.

The relative stability of cluster formation based on the reaction Gibbs free energy from the interaction among CA, SA, AM, DMA and W molecules are depicted in Fig. 5. The results show that SA·DMA is the most stable dimer, (SA)$_2$·DMA is




the most stable trimer followed by SA·DMA·CA, as well as (SA)$_2$·DMA·CA is the most stable tetramer followed by (SA)$_3$·DMA.

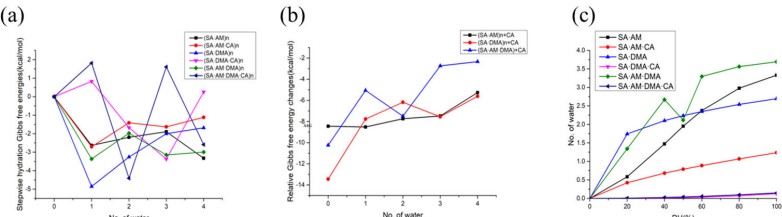

**Figure 4: Calculation of (a) stepwise hydration free energy; (b) reactive Gibbs free energy change values (c) average hydration number of SA·AM·W$_n$·CA$_m$, SA·DMA·W$_n$·CA$_m$ and SA·AM·DMA·W$_n$·CA$_m$ (n = 0 - 4, m=0-1) clusters, using the M06-2X/6-311+G(2d, p) level at T = 298.15 K and p = 1 atm state.**

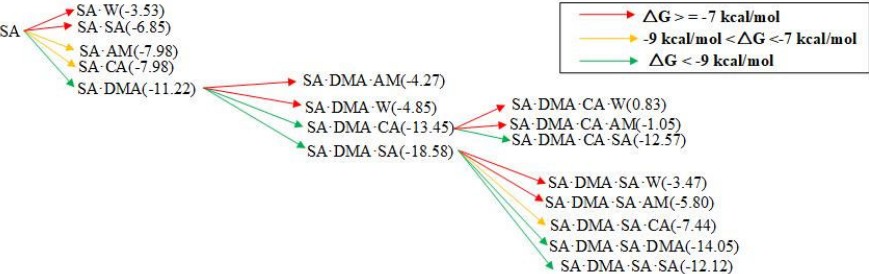

**Figure 5: Possible pathways for clusters formation based on reaction Gibbs free energy.**

### 3.3 Hydrate distribution

In this study, the hydration distributions of SA·AM·W$_n$·CA$_m$, SA·DMA·W$_n$·CA$_m$ and SA·AM·DMA·W$_n$·CA$_m$ (n = 0 - 4, m=0-

1) clusters were calculated for three typical atmospheric relative humidities (RH=20%, 50%, and 80%) at a temperature of 298.15 K, as illustrated in Figure 6.

The hydration distribution varies significantly among different clusters. For SA·AM·W$_n$ (n = 0 - 4) clusters, the hydrate distribution is highly sensitive to RH. At low RH (20%), anhydrous SA·AM clusters are predominantly present ($x_0$ = 57.9%), followed by monohydrate SA·AM·W ($x_1$ = 31.0%). At medium RH (50%), the hydrate distribution becomes more uniform:

with percentages of approximately 28.4%, 27.3%, 20.4%, and 17.3% for SA·AM·W$_4$ cluster, SA·AM·W, SA·AM, and SA·AM·W$_2$, respectively; while only a small percentage of 6.6% corresponds to the presence of SA·AM·W$_3$ cluster. At high RH (80%), the predominant form is represented by SA·AM·W$_4$ clusters ($x_4$=57.9%). However, the introduction of a CA molecule within the SA·AM·W$_n$ (n = 0 - 4) cluster leads to distinct changes in the hydrate distribution. In all three cases of RH, the predominant forms remain anhydrous and monohydrate are the present forms, i.e., the percentage of anhydrous

SA·AM clusters and monohydrate SA·AM·W with percentages of 60.6% and 36.7% at RH=20% respectively, and 35.2% and



53.3% at RH=50% respectively. Furthermore, in case of high RH (80%), the percentages are 22.8% for anhydrous SA·AM clusters and 55.3% for monohydrates SA·AM·W.

For SA·DMA·$W_n$ (n = 0 - 4) clusters, SA·DMA·$W_2$ and SA·DMA·W clusters were the predominant forms at low relative humidity (RH = 20%), with 53.2% and 34.5%, respectively. While SA·DMA·$W_2$ and SA·DMA·$W_3$ clusters are the predominantly present forms (54.3% and 24.8% respectively in RH=50 and 45.3% and 33.1% respectively in RH=80%). Interestingly, SA·DMA accounted for no more than 1.5% in all three relative humidity cases, which is negligible. More interestingly, the addition of a CA molecule within the SA·DMA·$W_n$ (n = 0 - 4) cluster results in a very different hydrate distribution, i.e., anhydrous clusters are the predominant form with a percentage of more than 97.2% in all three relative humidity cases.

For SA·AM·DMA·$W_n$ (n = 0 - 4) clusters, SA·AM·DMA·W and SA·AM·DMA clusters were the predominantly present forms at low relative humidity (RH = 20%), with percentages of 46.3% and 25.1%, respectively. However, SA·AM·DMA·$W_4$ and SA·AM·DMA·$W_3$ clusters are the predominantly existent forms (52.9% and 21.5%, respectively in RH=50%, 72.2% and 18.4% respectively in RH=80%). Interestingly, the addition of a CA molecule within the SA·AM·DMA·$W_n$ (n = 0 - 4) clusters resulted in a very different hydrate distribution, i.e., anhydrous clusters were the predominant form with a percentage of more than 95.1% in all three relative humidity cases.

From the above discussion, it can be seen that the addition of a CA molecule to the SA·AM·$W_n$ (n = 0 - 4), SA·DMA·$W_n$ (n = 0 - 4) and SA·AM·DMA·$W_n$ (n = 0 - 4) clusters resulted in a significant change in the hydrate distribution, i.e., in all three RH scenarios, anhydrous and monohydrate in the SA·AM·CA·$W_n$ (n = 0 - 4) clusters were the predominant forms present, and anhydrous clusters were the predominant forms present in SA·DMA·CA·$W_n$ (n = 0 - 4) and SA·AM·DMA·CA·$W_n$ (n = 0 - 4) clusters. Hence, addition of CA molecule considerably lowers the hydrophilicity of the SA·DMA·$W_n$ (n = 0 - 4) and SA·AM·DMA·$W_n$ (n = 0 - 4) clusters.

The hydration curves are shown in Fig. 4(3), where the maximum number of water molecules of SA·AM·Wn (n = 0 - 4), SA·DMA·Wn (n = 0 - 4), SA·AM·DMA·Wn (n = 0 - 4) clusters were 3.33, 2.70, and 3.69, respectively, 3.33, 2.70, and 3.69, respectively when the RH was close to 100%. While adding one CA molecule to the three above clusters, the maximum number of water molecules decreased to 1.24, 0.13 and 0.13, further indicating that the addition of CA significantly reduced the hydrophilicity of the clusters, which is consistent with the above conclusion.



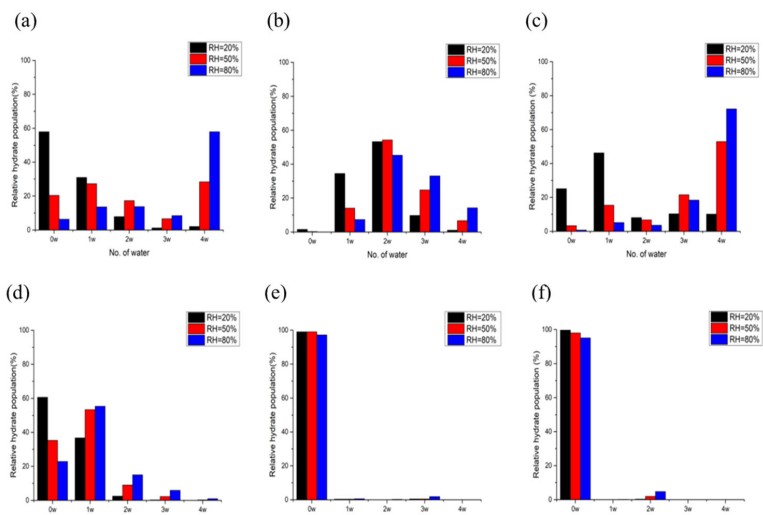

**Figure 6: Hydration distribution of (1) $SA \cdot AM \cdot W_n$ (n = 0 - 4), (2) $SA \cdot DMA \cdot W_n$ (n = 0 - 4), (3) $SA \cdot AM \cdot DMA \cdot W_n$ (n = 0 - 4), (4) $SA \cdot AM \cdot CA \cdot W_n$ (n = 0 - 4), (5) $SA \cdot DMA \cdot CA \cdot W_n$ (n = 0 - 4) and (6) $SA \cdot AM \cdot DMA \cdot CA \cdot W_n$ (n = 0 - 4) clusters at 298.15 K with different**
**RH (20%, 50%, and 80%), respectively.**

### 4 Conclusions

The role of a citric acid (CA) molecule in the nucleation of $SA \cdot AM \cdot W_n \cdot CA_m$, $SA \cdot DMA \cdot W_n \cdot CA_m$, and $SA \cdot AM \cdot DMA \cdot W_n \cdot CA_m$ (m=0-1, n = 0 - 4) clusters is investigated in this study using the M06-2X/6-311+G(2d, p) level. Our findings can be summarized as follows:

325       Structurally, CA's two α-COOH groups, one β-COOH group, and one β-OH group act as versatile hydrogen donors and acceptors. They actively participate in hydrogen bond formation, undergo proton transfer phenomena, contribute to the formation of a cage-like structure that effectively lowers nucleation barriers, and play a crucial role in stabilizing the $SA \cdot AM \cdot W_n$, $SA \cdot DMA \cdot W_n$, and $SA \cdot AM \cdot DMA \cdot W_n$ (n = 0 - 4) clusters.

      Thermodynamically, the addition of a citric acid (CA) molecule to $SA \cdot AM \cdot W_n$, $SA \cdot DMA \cdot W_n$, and $SA \cdot AM \cdot DMA \cdot W_n$ (n
= 0 - 4) clusters resulted in negative Gibbs free energies for all reactions. According to the second law of thermodynamics, it is further demonstrated that the inclusion of citric acid promotes cluster formation and significantly contributes to sulfuric acid-base-water nucleation for new particle formation.

      Hydaratingly, incorporating a CA molecule into $SA \cdot AM \cdot W_n$, $SA \cdot DMA \cdot W_n$, and $SA \cdot AM \cdot DMA \cdot W_n$ (n = 0 - 4) clusters led to a pronounced alteration in hydrate distribution. Specifically, at relative humidities of 20%, 50% and 80%, anhydrous
and monohydrate forms of $SA \cdot AM \cdot CA \cdot W_n$ (n = 0 - 4) were predominant within the clusters while anhydrous clusters prevailed



in both $SA \cdot DMA \cdot CA \cdot W_n$ (n = 0 - 4) and $SA \cdot AM \cdot DMA \cdot CA \cdot W_n$ (n = 0 - 4) clusters. Consequently, it can be inferred that citric acid diminishes the hydrophilicity of these three types of clusters.

The involvement of citric acid in the formation of sulfuric acid-base-water clusters through multi-component nucleation is evident from the aforementioned three aspects. In conclusion, the presence of citric acid in the atmosphere positively

influences cluster formation and facilitates multicomponent nucleation of sulfuric acid-base-water clusters. This study significantly contributes to our understanding of how citric acid participates in and enhances new particle formation events.

### Author contributions

Xiaoli Gong performed the calculations, analyzed the data; Renyi Zhang helped to interpret the results and revised the manuscript; Xiaoli Gong and Liyao Zhu prepared the picture and wrote the paper. All authors contributed to the manuscript

preparation and approved to the final version of the manuscript.

### Conflicts of interest

There are no conflicts to declare.

### Acknowledgements

The research was conducted with the advanced computing resources provided by Texas A & M High Performance Research

Computing.

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
