# Peer review of "Technical note: A theoretical study on the mechanism of citric aciddriven multi-component nucleation of sulfuric acid-base-water clusters"

_EGUsphere, 2023_

## Referee Comment (RC1)

**Review of Gong et al for Atmospheric Chemistry and Physics**

Gong and co-workers study clusters containing sulfuric acid (SA), ammonia (AM), dimethylamine (DMA), citric acid (CA) and water (W). Specifically, the $(SA)_1(AM)_1(CA)_1(W)_{0-4}$, $(SA)_1(DMA)_1(CA)_1(W)_{0-4}$ and $(SA)_1(AM)_1(DMA)_1(CA)_1(W)_{0-4}$ clusters are investigated using quantum chemical methods. The thermochemistry of the clusters are calculated using density functional theory at the M06-2X/6-311+G(2d,p) level. The hydrogen bonding patterns of the cluster structures are discussed in detail. The calculated thermochemistry is utilized to calculate the hydrate distributions of the clusters. The overall conclusion is that citric acid is important for cluster formation.

However, I do not find much direct evidence in the paper to adequately support this claim. The authors need to provide definitive evidence that citric acid actually binds and sticks to the clusters. The sign of the calculated standard binding free energy is not definitive proof, as it does not consider the stability of the clusters or atmospheric concentrations of the participating vapour. So please go beyond just reporting the standard free energies, as this would take the study to a new level. To provide more concrete evidence try calculating the evaporation rates, cluster equilibrium concentration at relevant atmospheric conditions or free energies at given conditions. To be absolutely certain larger clusters than considered here would also be required.

The abstract and conclusions appears to exaggerate the findings, giving the impression that citric acid is proved to be very important for nucleation. I would be very careful with such grandiose statements, as unknowing readers could take this at face value and misuse the data.

The applied M06-2X/6-311+G(2d,p) level also appears as an odd choice and should be further justified. Why this basis set? For such small clusters as those studied here the 6-311++G(3df,3pd) basis set, which for many groups are the standard for DFT calculations on clusters, is applicable.

On a broader level, I am missing a clear motivation to what research questions the authors are trying to answer in the context of atmospheric chemistry. Is it whether citric acid binds to the SA-AM, SA-DMA clusters? Is it the water uptake of the cluster? The presentation is not very clear, which makes it difficult to comprehend the atmospheric relevance of the study.

Overall, while the topic is of interest to the readership of *Atmospheric Chemistry and Physics*, I cannot recommend publication in *ACP* in its current form, as the direct relevance for atmospheric chemistry is not clearly conveyed. I suggest the manuscript is submitted to a journal that focus on general physical chemistry instead. Some detailed comments are given below.

**Comments**

**Line 11**: *"It has been shown that citric acid, present in atmosphere, is a potential precursor of new particles …"*

Are there actually any measurements of citric acid in the gas-phase? What is the gas-phase concentration?

**Line 13**: *"The thermodynamically stable geometry …"*

Please clarify what you mean by "thermodynamically stable geometry" here.

**Line 17**: *"This interaction lowers the nucleation barriers (⊿G< 0), indicating an energetically favorable reaction. "*

The quantum chemical calculations give you the free energy change, also known as the binding free energy. This concept is very different from the nucleation free energy barrier, which is a kinetic concept. Hence, to connect the two you need to calculate the free energies at given conditions. I.e. also include the vapour concentrations of the nucleating species.

**Line 18**: *"At three relative humidities (RH), …"*

Please explicitly state which three relative humidities you studied here.

**Line 20**: *"In conclusion, the involvement of citric acid in atmospheric processes is conductive to cluster formation, thereby facilitating the multicomponent nucleation of sulfuric acid-base-water clusters. Overall, our study highlights how citric acid participates in and enhances new particle formation processes. "*

I do not find any clear conclusions in the manuscript that citric acid is participating in and enhancing the new particle formation process. Be careful with just stating this as facts, without backing it up. It can lead to unknowing readers to misuse the data. Please explicitly state which of your results that support this conclusion.

**Line 29**: *"Currently, new particle formation events are frequently observed across various atmospheric environments worldwide, and understanding the nucleation mechanism of new particles in the atmosphere, particularly the thermodynamic stability of initial molecular clusters, has emerged as a prominent research topic within international atmospheric chemistry (Yao et al., 2018)."*

I do not understand how the reference from Yao et al fits into this sentence.

**Line 37**: *"New particles exhibit a wide range of shapes and sizes, …"*

Please supply a reference for this statement.

**Line 55**: *"… 3-methyl-1,2,3-butanetricarboxylic acid …"*

Actually, I believe that MBTCA was first studied by Ortega et al. (https://doi.org/10.1021/acs.jpca.5b07427). So please cite the original reference.

Subsequently our group also worked on electrically neutral (DOI: 10.1039/C6CP08127D) and ionic (https://doi.org/10.1021/acs.jpca.7b03981) MBTCA clusters. But the findings were that MBTCA could not form clusters together with sulfuric acid, but would rather form cluster by itself (https://doi.org/10.1021/acs.jpca.9b00428). No need to cite these papers, just for future reference.

**Line 64**: *"… is one of highly oxygenated multifunctional organic molecules (HOMs) …"*

I would not classify citric acid as a HOM, if you follow the definition laid out by Bianchi et al. (https://doi.org/10.1021/acs.chemrev.8b00395).

**Line 73**: *"… on the basis of a full-domain potential energy surface search algorithm, combined with a high-precision quantum chemical calculation method and the following three calculations are performed at the M06-2X/6- 311+G(2d,p) level in this paper "*

What do you mean by full-domain potential energy surface search algorithm? By looking at the methods section it appears that a random structure search has been employed.

I am also not completely comfortable to classify DFT calculations at the M06-2X/6-311+G(2d,p) level as "high precision".

**Section 2.1:** I am missing some comments on the accuracy of the applied configurational sampling methodology.

1) Some more information about how the "genmer" tool works in the Molclus package is warranted. Is this just a purely random structure search? How are the random structures generated? Are they initially optimized with a force field, or do you go straight to PM6?
2) Were the monomers during this initial sampling kept fixed? If citric acid is "locked" in a rigid local minimum structure, it might be difficult to rotate around the bonds and find the lowest free energy structure. Hence, this could be why one of the citric acid carboxylic acid groups is pointing away from the cluster in all the structures.
3) Only selecting the lowest 100 cluster configurations based on PM6 could lead to the global minimum cluster being missed (see Kurfman et al., https://doi.org/10.1021/acs.jpca.1c00872). Could the authors comment on this aspect?
4) How accurate are the M06-2X/6-311+G(2d,p) calculations? From Table 1 it appears that the calculated values differ significantly from the M06-2X/6-311++G(3df,3dp) calculations. This could indicate that the applied 6-311+G(2d,p) basis set is not sufficiently close to the basis set limit for yielding reliable results. Hence, the M06-2X/6-311+G(2d,p) values appears to lead to too strongly bound clusters.

**Line 112**: *"In order to obtain a global minimum structure on the potential energy surface, which is a steady state, …"*

This is not a steady state condition. Nothing is changing in time. It is sometimes called an equilibrium structure, but that is a different concept.

**Line 114**: *"Dispersion corrections being applied, …"*

What dispersion correction was applied? Grimmes empirical D3 dispersion?

**Line 125-127**: This is usually referred to as the binding free energy. Please correct the term throughout the manuscript, as it is otherwise quite easy to confuse it with stepwise reaction free energies.

**Line 141**: *"We will discuss the optimized global minimum energy …"*

It is always dangerous to claim to have found the global minimum. Essentially, it is impossible to know whether it is the global minimum or just a local minimum. Perhaps just rephrase it to the lowest identified minimum.

**Line 157-162**: *"… contains a strong hydrogen bond, a medium hydrogen bond, and a relatively weak hydrogen bond with bond lengths of 1.563 Å, 1.719 Å, and 2.067 Å respectively. "*

How do you classify the hydrogen bond strength here and in line 175-181 and 233-235? Is it just classified by the H-bond length and how robust is this classification? How does this information shows us something that cannot be deduced by calculated free energies? Please elaborate.

**Line 163**: *"When a citric acid molecule is added to SA·AM·W$_n$ (n = 0 - 4) …"*

The SA-AM-W clusters have been extensively studied in the literature. Are these calculated here or taken from somewhere else and then re-optimized? If re-calculated here, how does these clusters compare to the previously published structures.

**Figure 1-3**:

- Please add the level of theory in the figure captions.
- Did you find any conformers where all three carboxylic acid groups of citric acid are participating in the cluster formation process? If not, how much higher in free energy is that configuration. I find it curious that one -COOH group is dangling and not participating in hydrogen bonding in all the clusters. Is this an artefact of the simulation conditions? I.e. did you keep the monomers as rigid during the random structure search?

**Line 246**: *"… are all involved in the formation of the hydrogen bonds, which reduces the nucleation barrier, …"*

Could you please supply a reference to the statement that the number of hydrogen bonds definitively reduce the nucleation barrier? Does that not depend on the nature of those H-bonds?

**Section 3.1**: I am missing some clear indications as to what conditions the studied clusters are actually formed. Do the clusters have any appreciable concentration at realistic monomer concentrations and ambient conditions? What is the evaporation rate of the citric acid from the clusters. Does it actually stick and facilitate the further uptake of vapours?

**Figure 4,5 and 6**: Please increase the size and resolution of the figures. They are very hard to see and quite grainy.

**Line 261**: Judging from the values I suspect that these are the binding free energies as defined by your reaction (3) and not the reaction (stepwise) free energies? Please make this clearer.

**Line 262**: *"While addition of a CA molecule to SA·DMA·W$_n$ and SA·AM·DMA·W$_n$ (n = 0 - 4) clusters, only the first hydration reaction Gibbs free energies (-13.45 kcal mol$^{-1}$ and -10.24 kcal mol$^{-1}$) are more negative."*

This sentence is a bit difficult to comprehend. Do you mean the addition free energies of CA to the clusters or the binding free energy of clusters that contain CA? Also, the discussion here revolves around the absolute binding free energies. This is not necessarily the most useful quantity to discuss. It would be much more beneficial to discuss the addition free energy of CA or water to the clusters.

**Line 266**: *"It is clear that citric acid facilitates the stabilization of the clusters in this paper, and furthermore, the values of the Gibbs free energies of the reactions added to the multicomponent clusters are related to the degree of hydration as well as to the precursors. "*

Please clarify this sentence

- How do you judge stability? I guess a stable cluster will have negligible evaporation rate.
- How does citric acid facilitate the stabilization of the clusters?
- What do you mean by facilitate in this regard?

Judging from the free energies given in Figure 5, I would be more inclined to say that SA and DMA forms stable clusters. From the discussion it is difficult to judge whether CA actually participates.

**Figure 4**: Figure 4 (c) is not referenced in the text. It is referenced as figure 4(3) at line 312. Please correct.

**Section 3.3**: I am missing that the authors put the calculated hydrate distributions into the context of exiting literature. How does the hydrate distribution of the SA-AM and SA-DMA systems compare with previous work?

The fact that CA makes the clusters less hygroscopic is interesting. However, this is of course only relevant if CA actually bind strongly enough to the cluster.

**Line 325**: *"Structurally, CA's two α-COOH groups, one β-COOH group, and one β-OH group act as versatile hydrogen donors and acceptors. They actively participate in hydrogen bond formation, undergo proton transfer phenomena, contribute to the formation of a cage-like structure that effectively lowers nucleation barriers, and play a crucial role in stabilizing the SA·AM·$W_n$, SA·DMA·$W_n$, and SA·AM·DMA·$W_n$ (n = 0 - 4) clusters. "*

The structure itself cannot be used to directly infer that the nucleation barrier goes down. Please re-write.

**Line 329**: *"Thermodynamically, the addition of a citric acid (CA) molecule to SA·AM·$W_n$, SA·DMA·$W_n$, and SA·AM·DMA·$W_n$ (n = 0 - 4) clusters resulted in negative Gibbs free energies for all reactions. According to the second law of thermodynamics, it is further demonstrated that the inclusion of citric acid promotes cluster formation and significantly contributes to sulfuric acid-base-water nucleation for new particle formation."*

Please clarify how the second law of thermodynamics fit into this context.

**Line 328**: *"The involvement of citric acid in the formation of sulfuric acid-base-water clusters through multi-component nucleation is evident from the aforementioned three aspects. "*

Please explicitly explain how it is evident.